# A novel murine model of post-implantation malaria-induced preterm birth

**Alicer K. Andrew** [1,2], **Caitlin A. Cooper**[3], **Julie M. Moore**[1,2]*

**1** Department of Infectious Diseases and Immunology, College of Veterinary Medicine, University of Florida, Gainesville, Florida, United States of America, **2** Department of Infectious Diseases, College of Veterinary Medicine, University of Georgia, Athens, Georgia, United States of America, **3** Department of Cellular Biology, Franklin College of Arts and Sciences, University of Georgia, Athens, Georgia, United States of America

☯ These authors contributed equally to this work.
¤ Current address: Department of Infectious Diseases and Immunology, University of Florida, Gainesville, Florida, United States of America
* juliemoore@ufl.edu

**Data Availability Statement:** All relevant data are within the paper and its Supporting Information files.

## Abstract

Despite major advances made in malaria treatment and control over recent decades, the development of new models for studying disease pathogenesis remains a vital part of malaria research efforts. The study of malaria infection during pregnancy is particularly reliant on mouse models, as a means of circumventing many challenges and costs associated with pregnancy studies in endemic human populations. Here, we introduce a novel murine model that will further our understanding of how malaria infection affects pregnancy outcome. When C57BL/6J (B6) mice are infected with *Plasmodium chabaudi chabaudi* AS on either embryonic day (E) 6.5, 8.5, or 10.5, preterm birth occurs in all animals by E16.5, E17.5, or E18.5 respectively, with no evidence of intrauterine growth restriction. Despite having the same outcome, we found that the time to delivery, placental inflammatory and antioxidant transcript upregulation, and the relationships between parasitemia and transcript expression prior to preterm birth differed based on the embryonic day of infection. On the day before preterm delivery, E6.5 infected mice did not experience significant upregulation of the inflammatory or antioxidant gene transcripts examined; however, peripheral and placental parasitemia correlated positively with *Il1β*, *Cox1*, *Cat*, and *Hmox1* placental transcript abundance. E8.5 infected mice had elevated transcripts for *Ifnγ*, *Tnf*, *Il10*, *Cox1*, *Cox2*, *Sod1*, *Sod2*, *Cat*, and *Nrf2*, while *Sod3* was the only transcript that correlated with parasitemia. Finally, E10.5 infected mice had elevated transcripts for *Ifnγ* only, with a tendency for *Tnf* transcripts to correlate with peripheral parasitemia. Tumor necrosis factor deficient (TNF-/-) and TNF receptor 1 deficient (TNFR1-/-) mice infected on E8.5 experienced preterm birth at the same time as B6 controls. Further characterization of this model is necessary to discover the mechanism(s) and/or trigger(s) responsible for malaria-driven preterm birth caused by maternal infection during early pregnancy.

**Funding:** This work was supported by the 2017-2019 Peach State LSAMP Bridge to the Doctorate program at the University of Georgia (NSF award 1702361), the University of Georgia's Department of Infectious Diseases, the University of Florida's College of Veterinary Medicine, and a National Institute of Health grant (award R01HD46860). The funders had no role in study design, data collection and analysis, decision to publish, or preparation of the manuscript.

**Competing interests:** The authors have declared that no competing interests exist.

## Introduction

Malaria remains a global public health issue despite unprecedented successes in vector control programs and antimalarial treatment. In 2020, the World Health Organization (WHO) reported that in regions with moderate to high *Plasmodium falciparum* transmission, roughly 11 million pregnancies were exposed to malaria infection [1]. Among those pregnancies, an estimated 819,000 infants were born with low birth weight, a well-documented risk factor for neonatal mortality [1]. Additional maternal-fetal health consequences to infection include maternal anemia and preterm delivery, especially in primigravid women [2]. Notably, malaria infection during pregnancy can manifest as an organ-specific syndrome known as placental malaria (PM), identified by the accumulation of *Plasmodium*-infected red blood cells (IRBCs), immune cell infiltration, and both malaria pigment (hemozoin) and fibrin deposition in the placenta [3–5]. To protect pregnant women from PM, WHO recommends the use of insecticide-treated mosquito nets (ITNs) and intermittent preventative treatment using antimalarial drugs such as sulfadoxine-pyrimathamine (IPTp-SP). However, in 2018 only 31% of women received the recommended dosages of IPTp-SP during their pregnancy and ITN coverage in sub-Saharan Africa has stalled since 2015 [6].

Despite decades of discovery and billions of dollars invested in malaria research, the underlying mechanisms involved in PM pathogenesis are still incompletely understood. One major reason is that access to the placenta is limited until after delivery, making it difficult to evaluate the impact of infection on the placenta during earlier stages of pregnancy. As a result, most of what is known about human PM comes from studies focused on clinical outcomes throughout gestation, systemic evaluation for biomarkers of disease, and placental biopsies postpartum. Some studies provide evidence for the critical role of inflammation, specifically via the upregulation of tumor necrosis factor (TNF), interleukin-10 (IL-10), interleukin-1 beta (IL-1β), and interferon-gamma (IFN-γ), in driving poor pregnancy outcomes [7–13]. Other studies suggest that oxidative stress, resulting from a cell's inability to mount an effective antioxidant response to mitigate damage driven by reactive oxygen species (ROS), is a key player in the placental pathology and poor fetal health outcomes associated with PM [14, 15].

Mouse models have been central to furthering our understanding of PM pathogenesis by providing a more accessible and genetically tractable tool for recapitulating malaria infection during pregnancy. Infection with the murine-infective *Plasmodium chabaudi chabaudi* AS (*Pcc*) in C57BL/6J (B6) and Swiss Webster mice capture some of the important hallmarks of human PM, such as elevated proinflammatory cytokine production, accumulation of IRBCs in the placenta, increased fibrin deposition, and pregnancy loss [16–20]. In these models, mice infected with *Pcc* on the first day of gestation (embryonic day 0.5, E0.5) experience a non-lethal infection that is accompanied by severe maternal anemia, high parasite burden, elevated inflammatory responses, hemozoin accumulation, and lipid peroxidation in the placenta, a known sign of oxidative stress. Therapeutic interventions that target the pathogenic contributions of coagulation, TNF, and ROS in B6 mice protect against pregnancy loss at midgestation (E10.5–12.5) [5, 19, 20]. Alternatively, studies in later gestation commonly utilize *Plasmodium berghei* in either B6 or BALB/c mice; however, the infection must be initiated between E10.5–13.5, due to the maternal lethality of the parasite. In these models, mice experience elevated placental inflammation and fibrin deposition, oxidative stress, IRBC adherence to the placenta, and preterm delivery [21–24]. Moreover, antimalarial and anti-inflammatory drug treatment restore maternal survival, reduce oxidative stress, and improve pregnancy outcomes [22, 25].

While these well-established models are useful for studying PM pathogenesis, our understanding of how the timing of malaria infection determines maternal and fetal health outcomes remains unclear. Studies of women naturally exposed to malaria at various times during

gestation report associations between peripheral and placental parasitemia and poor birth outcomes such as low birth weight, preterm delivery, and neonatal mortality [26–29], with some studies that focus on infection early in the first trimester [30–33]. However, due to a lack of histological and pathophysiological information related to PM pathogenesis during the first and second trimesters of pregnancy, it is difficult to envision host-directed therapeutic interventions that mitigate the impact of infection during these times in gestation. This represents a major obstacle in PM prevention and treatment efforts, especially amidst emerging antimalarial drug and insecticide resistance [34, 35].

Here, we describe a mouse model for studying malaria infection during early stages of pregnancy (early post-implantation) using *Pcc* in B6 mice. Given the developmental similarities between the first trimester human placenta and the last two-third of mouse gestation [36, 37], this model offers an avenue for investigating the pathogenic mechanisms underlying malaria infection during early placentation, when *P. falciparum* prevalence in pregnant women is highest [2] and fetal health is significantly impacted [27, 30, 33] but the placenta remains inaccessible to study. This work shows that malaria infection during the post-implantation period induces preterm birth and alters the expression of inflammatory and antioxidant-related gene transcripts in the placenta, one day prior to preterm delivery.

## Materials and methods

### Mice

C57BL/6J (B6), TNFα-knockout (B6;129S-Tnftm1Gkl/J), TNFRp55-deficient (Tnfrsf1atm1-Mak/J), and A/J mice were purchased from the Jackson Laboratory (Bar Harbor, ME) and maintained by brother-sister mating for a maximum of ten generations in the University of Georgia Coverdell Vivarium, following guidelines and regulations set forth by the University of Georgia Animal Care and Use Committee. All animals were supplied food (PicoLab® Rodent Diet 20: 5030, St. Louis, MO) and water ad libitum. Mice were adjusted to a 14-hour light/10-hour dark cycle and housed in conditions of 65–75 ˚F and 40–60% humidity. All animal procedures reported in this study were reviewed and approved by the Institutional Animal Care Use Committee (IACUC) at the University of Georgia, protocol number A2018 02-016-Y1-A0. Mice were anesthetized with 2.5% Tribromoethanol before sacrifice and all efforts were made to minimize suffering.

### Parasites and infection monitoring

The following reagent was obtained through BEI Resource Repository, NIAID, NIH: *Plasmodium chabaudi chabaudi*, Strain AS, MR4-741, contributed by David Walliker. Parasites were maintained as frozen stock in accordance with supplier guidelines and passaged in A/J mice for the purposes of infecting experimental B6 mice. Peripheral parasitemia was assessed by flow cytometry with a method adapted from work published by Jimenez-Diaz et al. [38]. A 2μl blood sample was collected by tail clip [39], diluted in 98μl 0.9% NaCl and stained with 0.25μl SYTO-16 Green Fluorescent Nucleic Acid Stain (ThermoFisher Scientific, catalog # S7578) within 4 hours of collection. Stained samples were incubated in the dark for 20 minutes at room temperature, further diluted 1:9 in 0.9% NaCl, then analyzed using a CyAn ADP Flow Cytometer (Beckman Coulter; Brea, CA). 30,000 cells were assessed daily for each mouse; infected red blood cells were distinguished based upon size and fluorescence intensity. An uninfected blood sample was used as an internal negative control. Parasitemia is reported as the percentage of infected red blood cells (IRBCs) to the total number of red blood cells (RBCs).

## Experimental design

Female B6 mice aged 8–10 weeks were paired with age-matched males nightly and examined each morning until a vaginal plug was observed, indicating successful mating. The morning a vaginal plug was observed was considered embryonic day 0.5, E0.5. After baseline measurements of weight and hematocrit were recorded, females were left undisturbed until E6.5 to minimize stress and increase the chances of successful blastocyst implantation. Mice were infected intravenously with 1000 *P. chabaudi chabaudi* AS-iRBCs diluted in 200ul 1X phosphate-buffered saline (PBS) per 20 grams of body weight on E6.5, E8.5, or E10.5 and are termed infected pregnant (IP). Age-matched non-pregnant mice were infected similarly as infection controls (infected non-pregnant, INP). In another control group, uninfected pregnant (UP) mice were sham injected with 200ul PBS per 20 grams body weight on the same gestation days. Immediately prior to infection or sham infection, experimental animals were switched to a high-fat rodent chow (PicoLab Mouse Diet 20 5058; St. Louis, MO) suited for pregnant animals. Weight measurements were recorded on E0.5, E6.5, E8.5, E10.5, and E12.5-E18.5 for all groups, depending on when the infection was initiated, to assess pregnancy progress and allow mice to proceed to spontaneous delivery. Parasitemia and hematocrit (a measure of anemia) were monitored daily in the infected groups beginning five days postinfection to assess the development of infection.

In a second series of experiments, mice infected on E6.5, E8.5, and E10.5, and along with their UP controls, were sacrificed on E15.5, E16.5, and E17.5, respectively. Weight, hematocrit, and parasitemia were recorded as described above until euthanasia. Mice were anesthetized before sacrifice and placentae were collected and preserved for histological and quantitative real-time PCR (RT-qPCR) analysis. Plump, pink, well-vascularized pups that reflexively responded to touch were considered viable and their weights and placenta weights were collected. Pup viability data are summarized in S1 Table.

## Histology

Placentae collected at the time of sacrifice were fixed for 24 hours in 10% buffered formalin, processed, and paraffin-embedded. Placental sections 5μm in thickness were mounted to microscope slides and stained with hematoxylin and eosin (H&E) for histological analysis. H&E images were obtained using a Keyence BZ-X710 with BZ Analyzer software and the 100X oil objective. Sections were Giemsa-stained and parasite burden in the placenta was determined by counting at least 1,000 erythrocytes in maternal blood sinusoids of at least two or more different placentae per dam, as previously described [19].

## Gene expression by quantitative real-time PCR

Total RNA from mouse placentae collected on E15.5, E16.5, and E17.5 was isolated using Trizol Reagent (Ambion, Ref # 15596026) and a bead shaker (BeadBlaster 24, Benchmark Scientific, SKU: D2400) with a minimum of four placentae were pooled per dam. RNA was DNasetreated (Invitrogen, Ref # AM1906) and then reverse-transcribed with High-Capacity cDNA Reverse Transcription Kit (Applied Biosystems, Ref # 4368814). Relative transcript abundance for the genes of interest was quantified using PowerSYBR Green PCR Master Mix (Applied Biosystems, Cat # 4367659), the Roche LightCycler 96 Instrument (software version 1.01.01.0050), and the Mic qPCR cycler (Biomolecular systems, firmware version 2.25). Each sample was assayed in duplicate for target and housekeeping genes. Average Ct values of target genes were normalized to average Ct values of Ubc as the reference gene and relative transcript abundance of genes of interest was determined using the ΔΔCt method. Transcript expression

in individual mice is presented relative to the mean expression value in UP mice at E6.5. Details of primer sets are summarized in S2 Table.

## Statistics

All statistical analyses were performed using GraphPad Prism version 9.2.0 (GraphPad Software; La Jolla, California). All raw clinical data are presented as mean ± SEM. Error bars are not visible if they are shorter than the symbol's height. The area under the curve (AUC) of percent starting weight, hematocrit, and parasitemia was calculated for each mouse between E0.5 and E18.5, as appropriate. AUC for weight and hematocrit was compared between IP, INP, and UP mice infected on the same day and was analyzed using a Kruskal-Wallis test between IP and INP groups and IP and UP groups. AUC for parasitemia between IP and INP groups was compared using a two-tailed unpaired t-test with Welch's correction. RT-qPCR data were analyzed using an unpaired t-test with Welch's correction and presented as a scatterplot with a bar representing the mean. Parasitemia and transcript data for correlation analyses were log-transformed. P values less than or equal to 0.05 were considered statistically significant. Mixed linear models analysis (SAS 9.4) was used to estimate differences in fetal and placental weights in dams sacrificed on E15.5, E16.5, and E17.5, and their controls. Proportional analysis tested by chi-square was used to compare pup viability between IP and UP dams. Univariate and multivariate regression analyses were done in SAS 9.4.

## Results

In this observational study, *Pcc* blood-stage infection was initiated on E6.5, 8.5, or 10.5 in B6 mice. These time points coincide with a period of intense placental growth and vascularization in human pregnancy, which represents the initial point at which placental sequestration of *P. falciparum* is proposed to occur [40, 41]. Initial measurements of weight and hematocrit were taken on E0.5 and then mice were randomly assigned to one of six groups–E6.5 infected pregnant (IP), E8.5 IP, E10.5 IP, E6.5 uninfected pregnant (UP), E8.5 UP, and E10.5 UP, with the indicated embryonic day representing the day of infection for IP groups (Fig 1A). After the infection was initiated, daily measurements of weight and parasitemia were collected beginning 5 days post-infection (Fig 1B–1G). In each group, IP mice steadily gained weight alongside their UP controls until they experienced a sudden and precipitous decline in weight, suggestive of preterm pregnancy compromise (Fig 1B–1D). Euthanasia of these dams at E19 revealed empty uteri, implicating preterm birth as the likely cause of the observed weight loss. The time to preterm birth was accelerated the later in gestation that infection occurred such that E6.5-, E8.5-, and E10.5-infected dams began losing weight nine, eight, and seven days post-infection, respectively (Fig 1B–1D). UP mice continued gaining weight beyond E18.5, as expected during a normal pregnancy. Parasitemia was greater in IP dams compared to their infected nonpregnant (INP) counterparts (Fig 1E–1G), which was confirmed statistically through area under the curve (AUC) analysis, in all groups except the E8.5 infection group (S1 Fig). Corresponding with higher parasite burdens, AUC for hematocrit was significantly lower in E6.5 infected dams compared to UP controls (S1 Fig), likely due to the prolonged destruction of red blood cells caused by the advanced stage of infection compared to other infection groups [42]. AUC for hematocrit in the other infection groups, E8.5 and E10.5, was not statistically significantly different from UP controls (S1 Fig).

To investigate the effect of maternal infection on placental and pup health in this model, dams infected on E6.5, E8.5, or E10.5 were euthanized one day prior to expected preterm parturition on E15.5, E16.5, and E17.5 respectively (Fig 1B–1D, arrows indicate the day of euthanasia when placentae were collected for analysis). Daily weight, parasitemia, and hematocrit

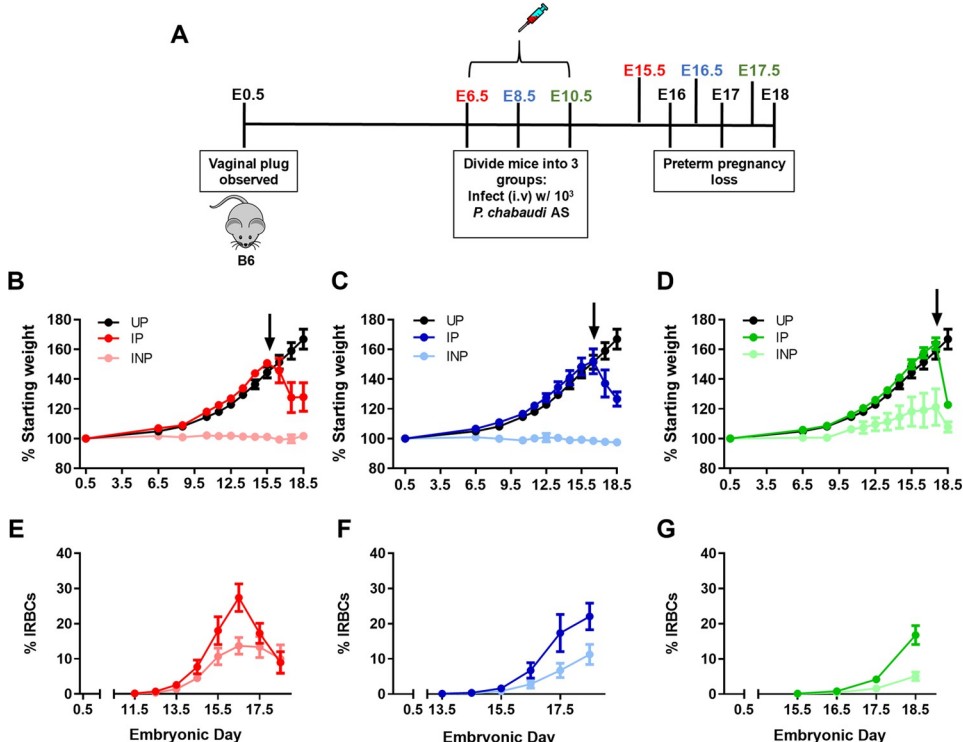

**Fig 1. *P. chabaudi* AS infection in the post-implantation period universally precipitates preterm delivery.** (A) Schematic of experimental design. (B-D) Percent starting weight and (E-G; % IRBCs) parasitemia are presented for infected pregnant (IP), uninfected pregnant (UP), and infected non-pregnant (INP) groups. All IP mice experienced precipitous weight loss during ascending infection. Arrows indicate timepoints chosen for subsequent experiments. (B, E) E6.5 infection group (red): IP, n = 6; UP, n = 6; and INP, n = 4. (C, F) E8.5 infection group (blue): IP, n = 8; UP, n = 6; INP, n = 4. (D, G) E10.5 infection group (green): IP, n = 14; UP, n = 6; INP, n = 7.

measurements were recorded, beginning 5 days post-infection (S2 and S3 Figs). Placental parasitemia in dams of the E8.5 and E10.5 infection groups, but not in those infected at E6.5, was significantly higher than their peripheral parasitemia at sacrifice (Fig 2). To further examine the impact of infection, pup and placental weights and pup viability were measured one day before expected preterm parturition. By mixed linear models analysis, neither pup nor placental weights were significantly impacted by maternal malaria across all infection groups; graphs are shown for visualization purposes only (Fig 3). Pup viability was also not different between IP versus UP dams, regardless of the infection group (S1 Table). When holding infection constant, increased pup viability in mice infected at E10.5 yielded a significant reduction in pup weight (by 0.7546 g, P = 0.0102), consistent with the well-documented observation that increased litter sizes result in reduced fetal weights [43]. Overall, these results indicate that despite having significant parasite burden, including in the placenta, malaria-induced preterm birth in this model occurs independently of changes in placental and fetal weight and viability.

Histology in PM-positive human placenta observed at term is generally characterized by immune cell infiltration, IRBC accumulation and both fibrin and hemozoin deposition in placental tissues [3–5]. To assess the histological condition of the placenta prior to preterm delivery in our model, placentae collected from IP mice and their UP controls were stained with hematoxylin and eosin (H&E) (Fig 4). Representative images of placentae from each group depict IRBCs in the placentae of all infection groups (Fig 4B, 4D and 4F). Giemsa-stained tissue sections further demonstrate the presence of IRBCs in all infection groups (S4 Fig). There

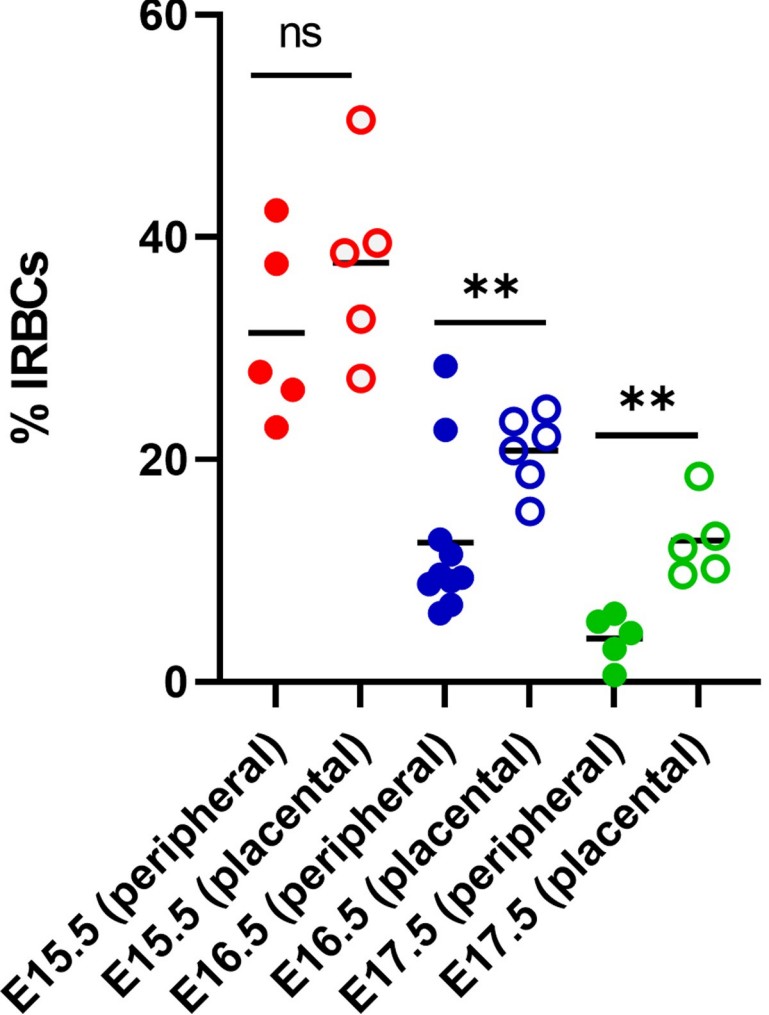

**Fig 2. *P. chabaudi* AS-infected red blood cells accumulate prior to preterm delivery in the placentae of mice infected at E8.5 and E10.5.** Peripheral and placental parasitemia from mice infected on E6.5 (red), E8.5 (blue) and E10.5 (green), sacrificed at E15.5, E16.5 and E17.5, respectively. **P < 0.002, unpaired t-test with Welch's correction; ns = not significant, P > 0.05.

was no evidence of immune cell accumulation or excessive fibrin deposition within the labyrinth in any of the placentae examined.

The pathogenesis of preterm delivery has long been associated with excessive inflammation, induced by infection or other disorders [44]. To probe the importance of inflammatory mediators in driving preterm delivery in these models, real-time quantitative PCR (RT-qPCR) analysis was performed using RNA isolated from the placental homogenates. The relative abundance of inflammatory gene transcripts, such as tumor necrosis factor (*Tnf*), interleukin-10 (*Il10*), interferon-gamma (*Ifnγ*), interleukin beta (*Il1β*), and genes important in pregnancy maintenance and parturition, such as cyclooxygenases 1 and 2 (*Cox1* and *Cox2*) were measured (Fig 5). Transcript abundance in placentae collected on E15.5 from mice infected on E6.5 revealed no significant differences between IP and UP dams for all inflammatory targets measured (Fig 5A–5F). However, *Il1β* transcripts were positively correlated with both peripheral parasitemia at euthanasia (E15.5) and peripheral parasitemia AUC for mice infected on E6.5 (Fig 6A and 6B). Additionally, there was a positive correlation between *Cox1* transcripts

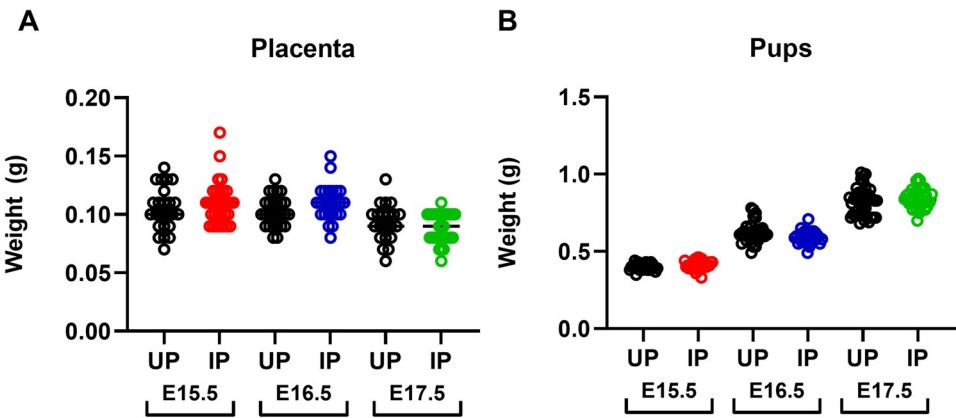

**Fig 3. *P. chabaudi* AS infection does not significantly impact placenta or pup weight prior to preterm delivery.**
(A) Placenta and (B) pup weights depicted for uninfected pregnant (UP) and infected pregnant (IP) mice sacrificed one day before preterm delivery. These data are shown for visualization only; statistical analysis is in the text. E15.5 group: 4 IP dams, 32 placentae; 4 UP dams, 30 placentae. E16.5 group: 4 IP dams, 31 placentae; 4 UP dams, 33 placentae. E17.5 group: 5 IP dams, 40 placentae; 5 UP dams, 41 placentae.

and peripheral parasitemia at sacrifice (Fig 6F) and a weak tendency toward a positive correlation between *Cox1* transcripts and parasitemia AUC for this same group (S5 Fig). In the E8.5 infection group, transcript abundance of all the inflammatory genes targeted, except for *Il1β*, was significantly elevated in placentae collected on E16.5 (Fig 5A–5F), but no correlative relationships with parasitemia at sacrifice or parasitemia AUC were found. In the E10.5 infection group, only transcripts for *Ifnγ* were elevated, while all other targets remained unchanged (Fig 5A–5F). There was a tendency for *Tnf* transcripts to be positively correlated with both peripheral parasitemia at sacrifice and parasitemia AUC in mice infected on E10.5 (S5 Fig).

To determine the parameters most critical in determining inflammatory gene expression, linear regression modeling was performed, considering infection status, day euthanasia was performed, and, as observed at euthanasia, uterus weight, number of embryos and embryo viability. The latter three were considered together in multivariate analysis, and none were found to predict transcript abundance. Infection status significantly influenced all targets (S3 Table). Day of euthanasia also significantly influenced all targets except *Cox1;* relative to placentae from mice infected at E6.5 (i.e., E15.5), transcripts for *Ifnγ, Tnf, Il1β, Il10, and Cox2* in placentae from E8.5 infections (i.e., E16.5) were all elevated. In a multivariate model testing both status and day, membership in the E16.5 group was a significant predictor for *Ifnγ, Tnf, Il1β, Il10, and Cox2* abundance (p ≤ 0.0306; S5 Table), while holding infection status constant. Similarly, while holding day constant, infection status maintained a significant influence on *Ifnγ, Il1β, Il10, and Cox2* abundance (p ≤ 0.0273), with tendencies for *Tnf* and *Cox1* (p = 0.0569 and p = 0.0645, respectively; S5 Table).

TNF has long been implicated as one of the major contributors to poor pregnancy outcome in malaria-positive primigravid women and in other infections [11, 45, 46]. Moreover, in a related mouse model, in which infection is initiated at E0.5, antibody-mediated ablation of TNF resulted in midgestational rescue from *Pcc*AS-induced abortion [19]. To directly evaluate a link between placental inflammation driven by TNF and subsequent precipitation of preterm delivery in the current mouse model, mice deficient in tumor necrosis factor (TNF$^{-/-}$) and tumor necrosis factor receptor 1 (TNFRI$^{-/-}$) were infected with 1000 *Pcc*-IRBCs on E8.5. This time point was chosen based on the multivariate modeling which suggested that *Tnf* transcripts are elevated prior to preterm delivery (after E16.5) in B6 mice. Contrary to expectation,

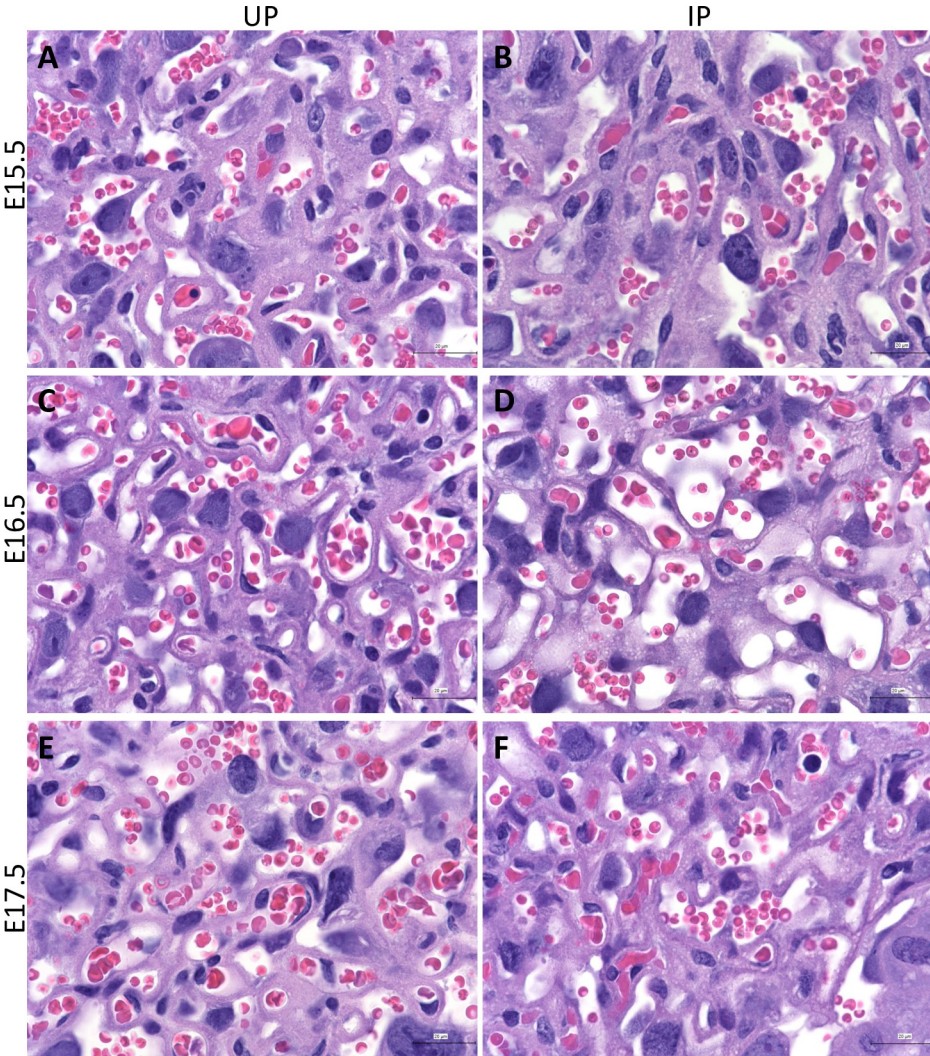

**Fig 4.** ***P.chabaudi*** **AS-infected red blood cells in the placental labyrinth region one day prior to preterm delivery.** (A, C, E) micrographs taken with 100x objective lens from uninfected pregnant (UP) controls at E15.5, E16.5, and E17.5. (B, D, F) micrographs from infected pregnant (IP) dams sacrificed one day prior to preterm delivery on E15.5, E16.5, and E17.5, respectively.

both TNF[-/-] and TNFRI[-/-] IP mice experienced preterm delivery after E16.5, similar to wild-type B6 mice (Fig 7). Although parasitemia appears to be higher in TNF[-/-] IP compared to B6 mice, AUC analysis revealed no significant differences between the groups for either TNF[-/-] or TNFRI[-/-] IP dams (S6 Fig). Taken together, these data demonstrate that TNF and TNF signaling through TNFRI are not required to drive preterm delivery in mice infected on E8.5.

Oxidative stress, generally described as an imbalance between the formation of free radicals, such as reactive oxygen species (ROS), and antioxidant defense molecules within cells and tissues, leads to cellular damage to lipids, proteins, RNA, and DNA. Recently, oxidative stress has been documented to play a crucial role in pregnancy outcome in some mouse models of PM [20, 23, 24] and is implicated in women experiencing malaria during pregnancy [14, 15] as well as other types of pregnancy complications [44, 47, 48]. To interrogate the role of oxidative stress in this model, placentae were evaluated for alterations in antioxidant gene expression by RT-qPCR (Fig 8). In the placentae of B6 mice infected on E8.5, placental transcript abundance at

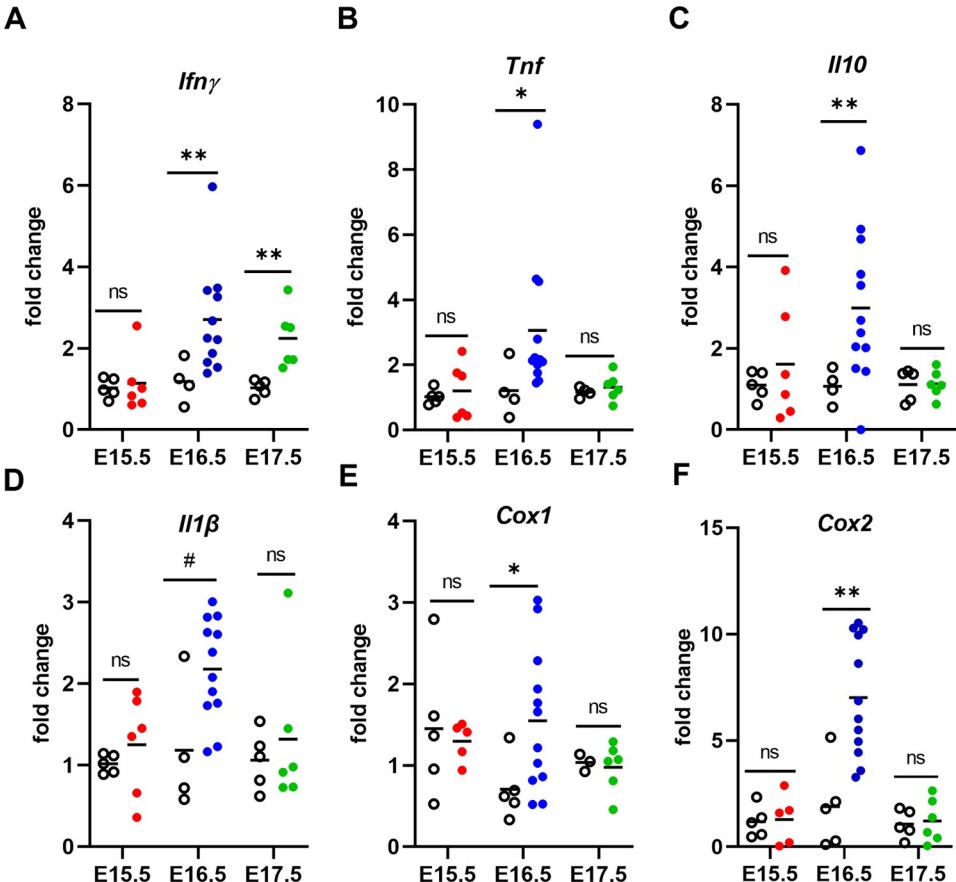

**Fig 5. *P. chabaudi* infection at E8.5 induces upregulation of inflammatory and parturition-associated gene transcripts in the E16.5 placenta prior to preterm delivery.** (A-F) Mouse *Ifnγ, Tnf, Il10, Il1β, Cox1,* and *Cox2* mRNA abundances normalized to *Ubc* and quantified by qPCR in placentae taken from infected pregnant (solid color circles) and uninfected pregnant (open black circles) dams. Solid color circles denote pooled placentae from individual infected dams sacrificed on E15.5 (red), E16.5 (blue), and E17.5 (green). Open black circles are pooled placentae from uninfected pregnant controls for each group. Group means and transcript abundance in individual mice are depicted. \*\*P ≤ 0.005, \*P < 0.05, unpaired t-test with Welch's correction; ns = not significant, P > 0.05; # denotes a trending result, P = 0.08.

E16.5 for superoxide dismutases 1 and 2 (*Sod1, Sod2*), catalase (*Cat*), and nuclear factor erythroid-2 related factor 2 (*Nrf2*) were significantly elevated compared to UP controls (Fig 7A, 7B, 7D and 7E). None of these targets were significantly elevated in IP mice infected and euthanized at the other time points (Fig 8). However, when the relationships between antioxidant transcript abundance and parasitemia were considered at E15.5 in the E6.5 infection group, *Hmox1* and *Cat* positively correlated with placental parasitemia. Additionally, *Hmox1* positively correlated with peripheral parasitemia AUC (Fig 6C–6E). In the E8.5 infection group, E16.5 placental *Sod3* transcripts tended to correlate negatively with both peripheral parasitemia at sacrifice and parasitemia AUC (Fig 9). Significant correlations between antioxidant transcript expression and parasite burden were not found in the placenta at E17.5 in the E10.5 infection group.

Linear regression modeling was applied to assess factors important in driving differences in placental antioxidant gene expression. As with inflammatory markers, this analysis considered infection status, day euthanasia was performed, uterus weight, number of embryos and embryo viability. The latter three were considered together and did not predict antioxidant transcript abundance. Infection status drove significant increases for genes except *Sod3* (p ≤ 0.04; S4

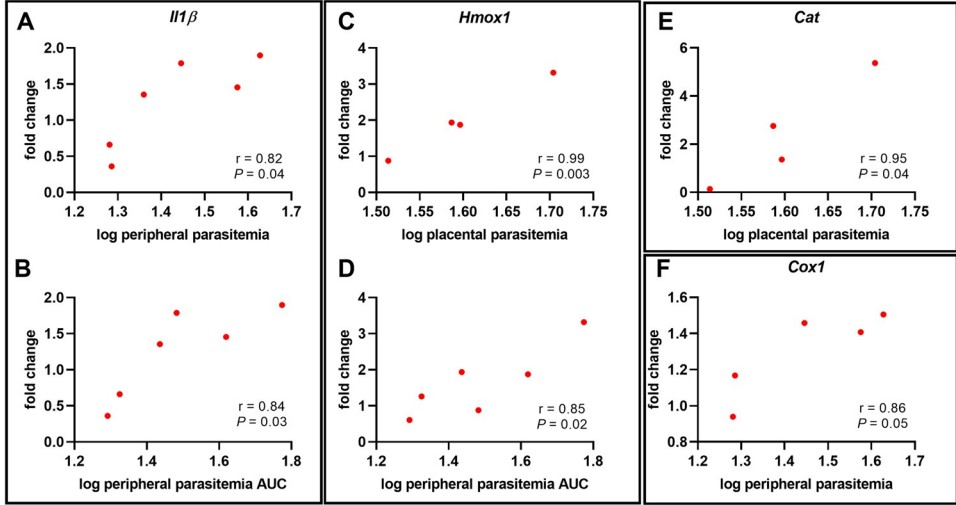

**Fig 6. Inflammatory and antioxidant gene expression in the E15.5 placenta positively correlates with peripheral and placental parasitemia in mice infected on E6.5.** (A-F) Mouse mRNA transcript abundance relative to peripheral parasitemia at the time of sacrifice, peripheral parasitemia AUC, and placental parasitemia one day before expected preterm birth (E15.5).

Table). Day of euthanasia also significantly influenced all targets except *Sod 3* and *Hmox1*. Transcripts for *Nrf2*, *Sod1*, *Sod2*, and *Cat* in placentae from E8.5 infections (i.e., E16.5) were all significantly elevated relative to placentae from mice infected at E6.5 (i.e., E15.5; $p \leq 0.01$; S4 Table). In a multivariate model testing influence of both status and day on these transcripts, holding day constant revealed a significant impact of infection was retained for *Nrf2*, *Sod1*, *Sod2*, *Cat* and *Hmox1* ($p \leq 0.04$; S6 Table). Membership in the E16.5 group independently predicted enhanced expression of *Nrf2*, *Sod1*, *Sod2*, *and Cat* ($p \leq 0.0254$; S6 Table).

## Discussion

Mouse models represent an affordable and genetically manipulable tool for investigating the pathogenesis of PM, a syndrome responsible for significant maternal morbidity and poor

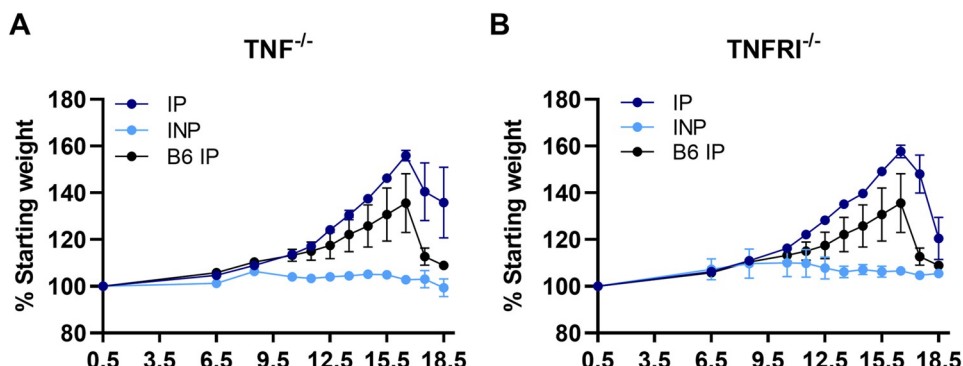

**Fig 7. Tumor necrosis factor (TNF$^{-/-}$) and tumor necrosis factor receptor 1 (TNFRI$^{-/-}$) deficient mice experience preterm delivery following *P. chabaudi* infection on E8.5.** (A) Percent starting weight for TNF$^{-/-}$ and (B) TNFRI$^{-/-}$ mice are depicted for infected pregnant (IP) and infected non-pregnant (INP) groups. TNF$^{-/-}$ IP: n = 3, INP n = 3; TNFRI$^{-/-}$ IP: n = 7, INP n = 3; B6 IP: n = 5. All IP mice experience precipitous weight loss, indicating preterm pregnancy compromise.

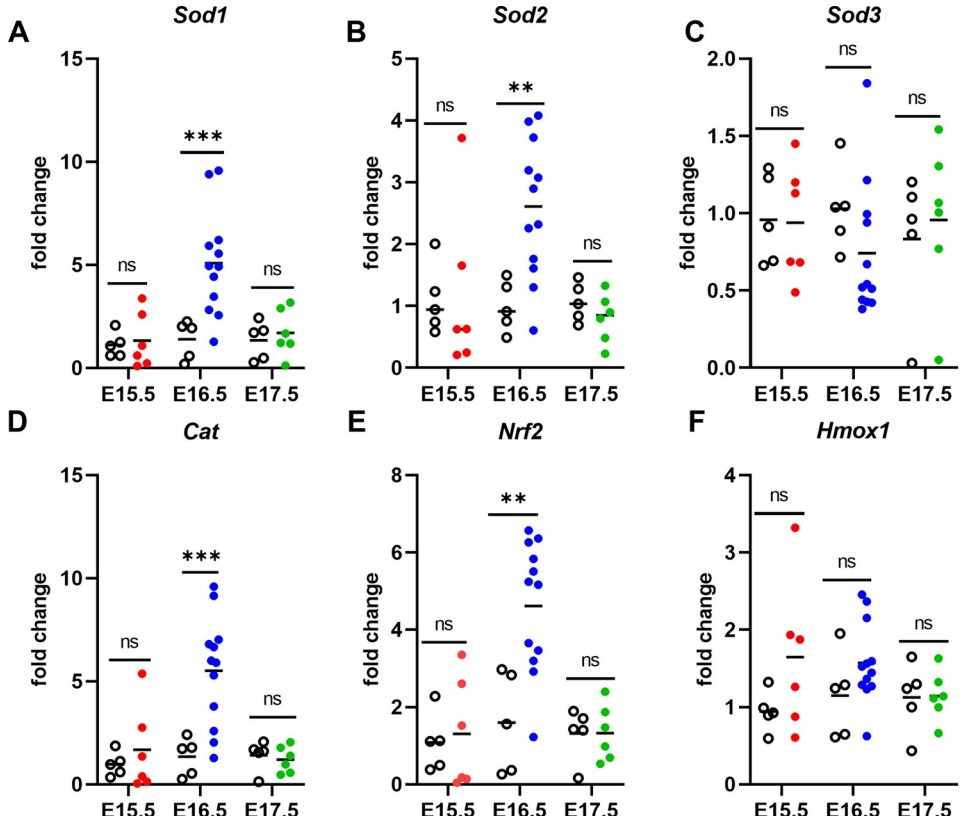

**Fig 8.** ***P. chabaudi* infection at E8.5 induces elevated antioxidant gene transcript expression in the E16.5 placenta prior to preterm delivery.** (A-F) Mouse *Sod1*, *Sod2*, *Sod3*, *Cat*, *Nrf2*, and *Hmox1* mRNA abundance relative to *Ubc* and quantified by qPCR in placentae taken from infected pregnant (solid circles) and uninfected pregnant (open black circles) dams. Solid color circles denote pooled placentae from individual infected dams sacrificed on E15.5 (red), E16.5 (blue), and E17.5 (green). Open black circles are pooled placentae from uninfected pregnant controls for each group. Group means and transcript abundance in individual mice are depicted. ***P ≤ 0.005, **P = 0.02; ns = not significant, P > 0.05.

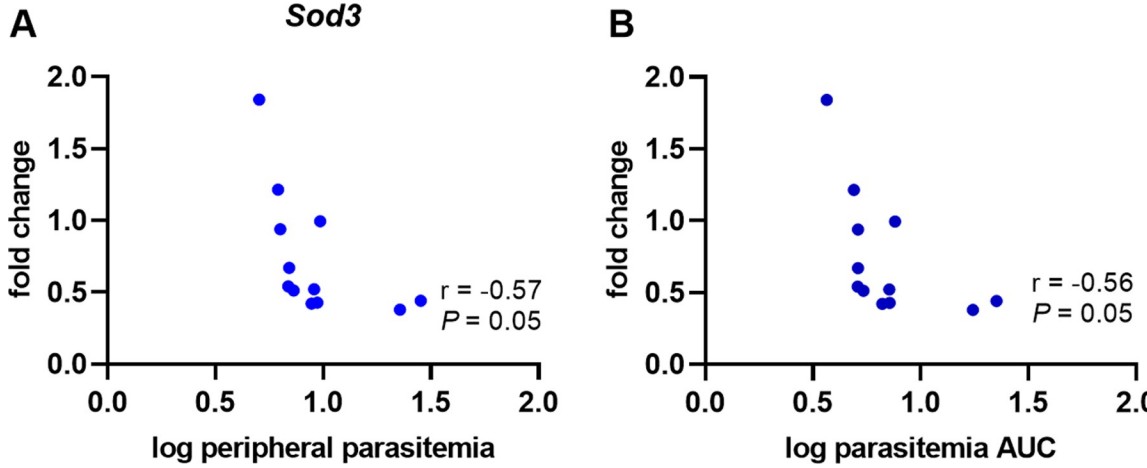

**Fig 9. Placental *Sod3* gene expression negatively correlates with peripheral parasitemia at sacrifice on E16.5 in mice infected on E8.5.** Mouse *Sod3* mRNA transcript abundance measured in placentae collected one day before expected preterm birth (E16.5) relative to (A) peripheral parasitemia at the time of sacrifice and (B) peripheral parasitemia AUC.

pregnancy outcomes globally. In this study, a novel murine model for malaria-induced preterm birth is described. This model requires a blood-stage infection in B6 mice using the murine infective parasite, *Plasmodium chabaudi chabaudi* AS (*Pcc*) at E6.5, E8.5, or E10.5. Although post-implantation *Pcc* infection in this model universally resulted in preterm birth, we observed that transcript abundance for certain genes in the placenta prior to delivery differed based on when in gestation the infection occurred. This model is the first to provide the flexibility to study malaria-induced pregnancy compromise driven by early post-implantation infection, without requiring acquired immunity or drug treatment for maternal survival. This is especially relevant to cases where immunity to PM is low or antimalarial treatment is not recommended, such as for primigravid women and those in their first trimester of pregnancy [13, 49].

To varying extents, key features of human PM are recapitulated in this model, including maternal anemia, higher parasitemia in the placenta compared to the periphery, and preterm delivery [4, 13]. Indeed, in the E8.5 and E10.5 infection groups, higher parasitemia in pregnant mice compared to non-pregnant controls corresponds with higher parasitemia in the placenta compared to peripheral parasitemia, corroborating previous studies in both mice and humans demonstrating that pregnancy itself can increase susceptibility to malaria [21, 24, 50] and lead to parasite accumulation in the placenta [3]. Thus, this novel model provides another opportunity for researchers to take a snapshot of the placenta at various times during the course of an acute malaria infection to investigate the relationships between parasite dynamics, placental development, and birth outcome.

Preterm delivery was a universal outcome of infection in the model reported here, with the time to delivery shortening the later in gestation the infection was introduced. When placental weight, pup weight, and pup viability were assessed one day prior to preterm delivery, the infection status of the dam had no impact. Thus, the physiological events that precipitate labor are acute. Analysis of the placenta preceding preterm delivery revealed differential expression of various inflammatory genes, highlighting the possibility that distinct mechanisms or triggers may be involved in driving preterm delivery in a gestational age-dependent manner in this model. Studies of cyclooxygenases, COX1 and COX2, have described their importance in the onset of labor in both mice and humans [51, 52]; however, *Cox1* and *Cox2* transcripts were elevated only at E16.5 in mice infected at E8.5. Interestingly, whereas *Cox1* was positively correlated with parasitemia in the E6.5 infection group, no significant malaria-induced upregulation for either *Cox1* or *Cox2* relative to uninfected placenta was detected in the E10.5 infection group at E17.5 or in the E6.5 infection at E15.5, both sampled one day prior to expected preterm delivery. Consistent with this, multivariate modeling showed that *Cox2* expression was influenced by infection only at E8.5, with a weak tendency for *Cox1* to be influenced by infection only.

IFNγ is elevated during acute malaria infection and required for parasite control [53]. *Ifnγ* transcripts were elevated only in the E8.5 and E10.5 infection groups even though all IP mice were experiencing ascending parasitemia at the time that placental analyses were conducted. Additionally, *Tnf* and *Il10* transcripts were increased in the E8.5 infection group only, and multivariate modeling confirmed that only infection at this day significantly predicted these increases. This was unexpected given the association of these cytokines with human malaria-induced preterm delivery [11, 54]. Likewise, TNF$^{-/-}$ and TNFR1$^{-/-}$ mice infected on E8.5 did not experience improved pregnancy outcomes or any changes in the time to preterm delivery. This was contrary to expectation given that ablation of TNF with antibody treatment successful protected mice from *Pcc*AS-induced abortion following infection at E0.5 [19]. These data suggest that although TNF signaling may be an important driver of poor pregnancy outcomes during human malaria infection [8, 19, 54] and early gestational malaria in the B6 mouse, its

expression at the placental level is not required within the 24 hours preceding preterm delivery after infection at E8.5. *Il1β* transcript abundance positively correlated with parasitemia in the E6.5 infection group and multivariate modeling showed that infection at E8.5 increased placental transcription of this factor. This observation is consistent with another model of malaria infection, in which reduction of IL-1β signaling improved pregnancy outcome [22]; thus, IL-1β appears to be a universal driver of poor birth outcomes in malaria, perhaps through inflammasome assembly and/or the initiation of pyroptosis [55].

Oxidative stress has been described in various models of preterm delivery [56] and implicated in both human and mouse PM [14, 15, 20, 23, 24]. To corroborate these observations, antioxidant gene transcripts were measured in placentae collected one day prior to expected preterm delivery. *Sod1*, *Sod2*, *Cat*, and *Nrf2* transcripts were elevated in the E8.5 infection group only, with *Hmox1* expression being predicted only by infection but not by infection day. Similar results were reported with induction of *Pcc*AS infection in B6 mice at E0.5, where *Sod1*, *Sod2*, *Cat*, and *Nrf2* transcripts were elevated in conceptuses and antioxidant drug treatment mitigated pregnancy loss [20]. When the relationship between parasite burden and antioxidant gene transcript expression was considered, *Cat* and *Hmox1* transcripts were positively correlated with placental parasitemia at sacrifice and *Hmox1* shared this relationship with peripheral parasitemia AUC as well. These results are consistent with the observed strong association between *Hmox1* expression and hemozoin deposition in the placentae of *Pcc*AS-infected Swiss Webster dams at midgestation [57]. Since E6.5 IP mice exhibited significant anemia and the greatest interval between infection and preterm delivery, infection initiated at this timepoint may present another opportunity to study the impact of *Hmox1* regulation of heme in driving negative pregnancy outcomes. Elevated heme levels and altered *Hmox1* activity have been implicated in the pathogenesis of both human and murine PM [58–60] and other pregnancy complications [61]. For mice infected on E10.5, none of the antioxidant gene transcripts measured were differentially expressed or correlated with parasite burden. Since this group had an overall lower parasite burden relative to the other groups, these results may collectively indicate that a threshold of infectious burden is required to commensurate certain antioxidant transcript responses in this model.

Our analyses focused on placental inflammation and oxidative stress; however, there are other mechanisms of infection-induced preterm delivery that were not assessed in this model. Some biological functions of clinical relevance in PM that could be explored in future studies include the coagulation pathway [5, 62], the complement system [63], nutrient bioavailability [64] and transport across the placenta [65, 66], angiogenic balance and vascularization [67], placental autophagy [68, 69], and hormonal alterations [65]. Likewise, it is possible that pathological changes leading to preterm birth occurred in tissue types or sites not evaluated here, such as the uteroplacental interface.

Given the many remaining unresolved questions in PM pathogenesis, the model described here may be instrumental for future studies aimed at modeling maternal and/or fetal responses to malaria infection during pregnancy. Although *Pcc* infection was initiated on three distinct embryonic days in this study, the system might collectively be employed as a single model for malaria-induced pregnancy compromise because the time points assessed (infection and sacrifice days) have been reported to share functional and developmental similarities with 1st and early 2nd trimester human placental development [36, 37]. Malaria infection in the 1st trimester is associated with detrimental effects on pregnancy [2, 32, 70] and represents a period in gestation where the placenta cannot be sampled and preventative antimalarial treatment is not recommended [49]. Detailed reports on the histopathological impact of malaria infection during the 1st and early 2nd trimester of human pregnancy, which correspond to the time points

modeled here, are currently unavailable. Thus, this model may provide a unique opportunity for researchers to gain insight into this underreported aspect of PM pathogenesis.

There are some limitations to this model and analysis as reported here. First, mouse gestation remains an imperfect model of human gestation, with a shorter gestational period and the birth of altricial young [37, 71–73]. However, the time points that we evaluate in the mouse reportedly share similarities with first trimester of human placentation [36, 37]. Second, this model did not yield evidence of intervillositis, placental fibrin deposits, intrauterine restriction, or low birth weight, all well-documented features of PM described in postpartum gestational tissues [4, 29, 33, 74], albiet at or close to term. Of note, histological characterization of the early placenta during malaria infection has not yet been reported; thus, our model helps to address this gap in knowledge. Third, it is possible that the timing of our analyses may have contributed to failure to detect critical physiological changes that occurred closer to the time of labor. Dams were euthanized at least 12 hours prior to expected preterm birth (which typically occurred overnight), and triggers for labor could have occurred inside this narrow window. Finally, the ability to successfully detect and characterize physiologically relevant changes by RT-qPCR alone may be limited. Thus, follow-up studies must consider expanding the timing of tissue collection and analyses, types of tissues evaluated, and the breadth of techniques used to identify a more precise cause(s) of preterm birth in this model.

In conclusion, this model provides a new avenue for interrogating the pathological driver (s) of preterm birth following post-implantation malaria infection. This model adds a new dimension to our current arsenal for studying the mechanisms involved in PM pathogenesis and could help improve our understanding of how malaria during pregnancy may galvanize preterm delivery.

## Supporting information

**S1 Fig. Area under the curve analysis for weight, parasitemia, and hematocrit in B6 mice infected with *P. chabaudi* AS on E6.5, E8.5, and E10.5.** (A-C) Area under the curve (AUC) was calculated for uninfected pregnant (UP), infected pregnant (IP) and infected non-pregnant (INP) mice belonging to the E6.5 infection group (A, D, G; red), E8.5 infection group (B, E, H; blue), or E10.5 infection group (C, F, I; green) in an observational study. No statistical differences are observed in weight change over time between UP and IP animals. (D-F) Parasitemia AUC is higher in some IP animals compared to INP counterparts. (G-I) Hematocrit AUC was statistically different in the E6.5 infection group only. Groups were compared either by using a Kruskal-Wallis test or unpaired t-test with Welch's correction (for parasitemia AUC). $^{**}P \leq 0.005$, $^{*}P < 0.05$; ns = not significant, $P > 0.05$.
(TIF)

**S2 Fig. Course of *P. chabaudi* AS infection in mice euthanized one day prior to expected pre-term delivery.** (A-C) Percent starting weight and (D-F) parasitemia (% IRBCs) are presented for infected pregnant (IP), uninfected pregnant (UP), and infected non-pregnant (INP) groups. All mice were sacrificed one day prior to expected preterm delivery (arrows indicate time of sacrifice) and tissues were collected for further studies. (A, D, G) E6.5 infection group, euthanized on E15.5 (red): IP, n = 9; UP, n = 9; and INP, n = 4. (B, E, H) E8.5 infection group, euthanized on E16.5 (blue): IP, n = 15; UP, n = 12; INP, n = 7. (C, F, I) E10.5 infection group, euthanized on E17.5 (green): IP, n = 12; UP, n = 10; INP, n = 6.
(TIF)

**S3 Fig. Area under the curve analysis for weight, hematocrit and parasitemia in mice euthanized one day prior to expected pre-term delivery.** Area under the curve (AUC) was

calculated for uninfected pregnant (UP), infected pregnant (IP) and infected non-pregnant (INP) mice belonging to the E6.5 infection group (A, D, G; red), E8.5 infection group (B, E, H; blue), or E10.5 infection group (C, F, I; green) for serial sacrifice experiments. (A-C) No statistical differences are observed in weight change over time between UP and IP animals. (D-F) Parasitemia AUC is higher in some IP animals compared to INP counterparts. (G-I) Hematocrit AUC achieved statistical significance between IP and INP mice in the E8.5 infection group only. Groups were compared either by using a Kruskal-Wallis test or unpaired t-test with Welch's correction (for parasitemia AUC). ****$P < 0.0001$, **$P < 0.005$, *$P < 0.05$; ns = not significant, $P > 0.05$.
(TIF)

**S4 Fig. *P.chabaudi* AS-infected red blood cells in the placental labyrinth region one day prior to preterm delivery.** (A-C) micrographs taken with 100x objective lens of Giemsa-stained placentae from infected pregnant (IP) dams sacrificed one day prior to preterm delivery on E15.5, E16.5, and E17.5, respectively.
(TIF)

**S5 Fig. Correlation analyses between inflammation and oxidative stress-associated transcripts and peripheral parasitemia in placentae of mice infected on E6.5 and E10.5.** Mouse mRNA transcript abundance relative to peripheral parasitemia at the time of sacrifice or parasitemia AUC in placenta collected one day before expected preterm birth. (A-B) *Tnf* transcripts tended to be correlated with parasitemia in the E10.5 infection group. (C-D) *Cat* and *Hmox1* transcripts tended to be positively correlated with parasitemia in the E6.5 infection group and (E) *Cox1* transcripts tended to have a positive correlation with parasitemia AUC.
(TIF)

**S6 Fig. Parasitemia and area under the curve for parasitemia in TNF$^{-/-}$ and TNFR1$^{-/-}$ mice infected on E8.5.** (A) Parasitemia (% IRBCs) in TNF$^{-/-}$ and (B) TNFRI$^{-/-}$ mice. (C-D) Area under the curve (AUC) analysis do not show a statistically significant increase in parasitemia between IP versus INP groups for both strains. TNF$^{-/-}$ IP: n = 3, INP n = 3; TNFRI$^{-/-}$ IP: n = 7, INP n = 3; ns = not significant, $P > 0.05$.
(TIF)

**S1 Table. Pup viability from infected and uninfected dams sacrificed one day prior to expected preterm delivery.** No differences were observed in pup viability between infected pregnant (IP) and uninfected pregnant (UP) dams across all infection groups. Statistical significance determined via proportional analysis tested by chi-square.
(DOCX)

**S2 Table. Primer sequences for qPCR targets.** Mouse-specific forward (FP) and reverse (RP) primers used in quantitative real-time PCR for the amplification of mRNA transcripts associated with inflammation, parturition, antioxidant activity, and reference (*Ubc*) genes.
(DOCX)

**S3 Table. Univariate regression analysis of inflammatory and parturition-associated transcript expression and day of sacrifice.** Analysis performed with proc reg for dichotomous (status) and continuous (parasitemia) and variables and proc glm for categorical variables (sacrifice day). Parasitemia was log10-transformed for the analysis. Dashes indicate that E15.5 is the reference value. Sample sizes for the analysis are as follows: E15.5 IP, n = 4; E15.5 UP, n = 4; E16.5 IP, n = 11; E16.5 UP, n = 4; E17.5 IP, n = 6; E17.5 UP, n = 3.
(DOCX)

**S4 Table. Univariate regression analysis of antioxidant transcript expression and day of sacrifice.** Analysis performed with proc reg for continuous (parasitemia) and dichotomous (status) variables and proc glm for categorical variables (sacrifice day). *Parasitemias were log10-transformed for the analysis. Dashes indicate that E15.5 is the reference value. Sample sizes for the analysis are as follows: E15.5 IP, n = 4; E15.5 UP, n = 4; E16.5 IP, n = 11; E16.5 UP, n = 4; E17.5 IP, n = 6; E17.5 UP, n = 3.
(DOCX)

**S5 Table. Multivariate regression analysis of inflammatory and parturition-associated transcript expression and day of sacrifice.** Analysis performed with proc glm. Dashes indicate that E15.5 is the reference value; dashes and NA indicate that these parameters were not considered in the analysis. Sample sizes for the analysis are as follows: E15.5 IP, n = 4; E16.5 IP, n = 11; E17.5 IP, n = 6.
(DOCX)

**S6 Table. Multivariate regression analysis of antioxidant transcript expression and day of sacrifice.** Analysis performed with proc glm. Dashes indicate that E15.5 is the reference value; dashes and NA indicate that these parameters were not considered in the analysis. Sample sizes for the analysis are as follows: E15.5 IP, n = 4; E16.5 IP, n = 11; E17.5 IP, n = 6.
(DOCX)

**S7 Table. Multivariate regression analysis of inflammatory and parturition-associated transcript expression and day of sacrifice.** Analysis performed with proc glm. Dashes indicate that E15.5 is the reference value; dashes and NA indicate that these parameters were not considered in the analysis. Sample sizes for the analysis are as follows: E15.5 IP, n = 4; E16.5 IP, n = 11; E17.5 IP, n = 6.
(DOCX)

**S8 Table. Multivariate regression analysis of antioxidant transcript expression and day of sacrifice.** Analysis performed with proc glm. Dashes indicate that E15.5 is the reference value; dashes and NA indicate that these parameters were not considered in the analysis. Sample sizes for the analysis are as follows: E15.5 IP, n = 4; E15.5 UP, n = 4; E16.5 IP, n = 11; E16.5 UP, n = 4; E17.5 IP, n = 6; E17.5 UP, n = 3.
(DOCX)

**S9 Table. Multivariate regression analysis of inflammatory and parturition-associated transcript expression and day of sacrifice.** Analysis performed with proc glm. Dashes indicate that E17.5 is the reference value; dashes and NA indicate that these parameters were not considered in the analysis. Sample sizes for the analysis are as follows: E15.5 IP, n = 2; E16.5 IP, n = 7; E17.5 IP, n = 3.
(DOCX)

**S10 Table. Multivariate regression analysis of antioxidant transcript expression and day of sacrifice.** Analysis performed with proc glm. Dashes indicate that E17.5 is the reference value; dashes and NA indicate that these parameters were not considered in the analysis. Sample sizes for the analysis are as follows: E15.5 IP, n = 2; E16.5 IP, n = 7; E17.5 IP, n = 3.
(DOCX)

**S1 Dataset.**
(XLSX)

## Acknowledgments

We thank MR4 for providing the malaria parasites contributed by David Walliker. We thank
Dr. Demba Sarr and Dr. Catherine D. Morffy Smith for providing the primers used for the
RT-qPCR assays. Alicer K. Andrew was supported by the 2017–2019 Peach State LSAMP
Bridge to the Doctorate Program at the University of Georgia (National Science Foundation,
Award # 1702361), the Department of Infectious Diseases at the University of Georgia, and the
College of Veterinary Medicine at the University of Florida. This work was supported by
National Institute of Health grants to J.M.M. The content is solely the responsibility of the
authors and does not necessarily represent official views of the *Eunice Kennedy Shriver*
National Institute of Child Health and Human Development, the National Institute of Allergy
and Infectious Diseases, or the National Institutes of Health.

## Author Contributions

**Conceptualization:** Caitlin A. Cooper, Julie M. Moore.

**Data curation:** Alicer K. Andrew, Caitlin A. Cooper.

**Formal analysis:** Alicer K. Andrew, Caitlin A. Cooper, Julie M. Moore.

**Funding acquisition:** Julie M. Moore.

**Methodology:** Caitlin A. Cooper.

**Resources:** Julie M. Moore.

**Software:** Julie M. Moore.

**Supervision:** Julie M. Moore.

**Visualization:** Alicer K. Andrew, Julie M. Moore.

**Writing – original draft:** Alicer K. Andrew.

**Writing – review & editing:** Alicer K. Andrew, Julie M. Moore.

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
