## [Decision Letter · Decision Letter 0]

16 Aug 2021

PONE-D-21-23277

Novel murine models of post-implantation and midgestional malaria-induced preterm birth

PLOS ONE

Dear Dr. Andrew,

Thank you for submitting your manuscript to PLOS ONE. After careful consideration, we feel that it has merit but does not fully meet PLOS ONE’s publication criteria as it currently stands. Therefore, we invite you to submit a revised version of the manuscript that addresses the points raised during the review process.

If you can address the specific concerns detailed below, then we would be willing to reconsider a revised version. Before seeing and evaluating such changes, however, we cannot guarantee that your revised article would be accepted for publication. Should you decide to revise the manuscript for further consideration here, your revisions should address all points indicated by reviewers.

We look forward to receiving your revised manuscript.

Kind regards,

Claudio Romero Farias Marinho, Ph.D.

Academic Editor

PLOS ONE

Journal Requirements:

https://journals.plos.org/plosone/s/file?id=ba62/PLOSOne_formatting_sample_title_authors_affiliations.

pdf

"A.K.A was supported by the 2017-2019 Peach State LSAMP Bridge to the Doctorate Program at the University of Georgia (National Science Foundation, award number 1702361, https://www.nsf.gov). This work was supported by a National Institute of Health grant to J.M.M (award number R01HD46860, www.nih.gov) . The content is solely the responsibility of the authors and does not necessarily represent official views of the Eunice Kennedy Shriver National Institute of Child Health and Human Development, the National Institute of Allergy and Infectious Diseases, or the National Institutes of Health."  

Reviewers' comments:

Reviewer's Responses to Questions

**Comments to the Author**

1. Is the manuscript technically sound, and do the data support the conclusions?

Reviewer #1: Yes

Reviewer #2: Yes

2. Has the statistical analysis been performed appropriately and rigorously? 

Reviewer #1: Yes

Reviewer #2: Yes

3. Have the authors made all data underlying the findings in their manuscript fully available?

Reviewer #1: Yes

Reviewer #2: Yes

4. Is the manuscript presented in an intelligible fashion and written in standard English?

Reviewer #1: Yes

Reviewer #2: Yes

5. Review Comments to the Author

Reviewer #1: In this manuscript, the authors describe three murine models of placental malaria, by infectingB6 mice with P. chabaudi, chabaudi at embryonic day 6.5, 8.5, or 10.5. The authors observed no intrauterine growth restriction, in contrast to other models of placental malaria, although they do report pre-term delivery. The authors also examined antioxidant and inflammatory gene transcripts, finding variations which are linked to the day of infection. Lastly, the authors have used mice deficient in TNF and its receptor to demonstrate that these do not drive pre-term delivery as a result of infection at embryonic day 8.5.

There are already a number of murine models of placental malaria, and the one described here provides an additional one which can be used to explore responses which result in pre-term delivery.

The statistical analysis methodology is appropriate considering the nature of the study and other recent work in the field.

There are, however, a few areas, which could be improved in the manuscript, listed below.

1. In lines 98 and 99, the authors state that infection during the second trimester is linked to poor outcomes of pregnancy and provide appropriate citations. They also claim that the mouse midgestation period (E6-12) is analogous to the human second trimester, but neither of the citations provided in that sentence support this. The human placenta is fully formed and exposed to maternal blood by the end of the first trimester, whereas the mouse placenta undergoes significant changes between days E6 and E12. These changes include chorioallantoic fusion around E8, and the fetuses are exposed to the full contents of maternal blood by E11.5-12. It is important that the authors clarify if they mean “the middle third of pregnancy” when calling the mouse midgestation and the human second trimester analogous, or if there are specific functional aspects.

2. Accounting for this series of changes in the mouse placenta during midgestation, it is pertinent for the authors to explain how the 3 models presented differ with respect to the phase of human disease they aim to recapitulate.

3. The authors use precipitous weight loss as an indicator of pre-term delivery (caption for figure 1 and lines 208-210), could this not also be an indicator for resorptions?

4. The histological data used to determine placental parasitemia in Figure 2 would benefit from the inclusion of a representative image showing the difference in maternal and fetal sinusoids in infected mice.

5. The caption for Figure 3 is difficult to follow e.g “…E6.5 infection group 4IP Dams, 32 placentae…”, Whereas the graph has labels corresponding to sacrifice date, and the number of data points do not appear to correspond (E15.5, 9 points).

6. The qPCR method states that pooled samples of minimum 4 placentas were used per dam (lines 167-169), whereas the caption of Figure 4 appears to indicate that each circle represents a placenta (rather than a pool) – this needs to be clarified.

7. For Figure 5, it is not clear why placental parasitemia (measured by histology and averaged, or gene expression in the same samples) was not used instead of, or in addition to maternal parasitemia.

8. In lines 369 to 372, the authors state that these are the first models they know of which do not induce maternal mortality. Models using recrudescence (Sharma et al 2016), specifically in multigravid mice (Marinho et al 2009) have also shown this.

9. The authors do not address the significant physiological differences in the human and mouse placenta within the manuscript, nor do they attempt to explore the potential limitations of their models. Adding this information to the discussion would allow the reader to determine whether the models are appropriate or not.

Reviewer #2: Major comments:

Introduction:

Overall, it was not clear in the introduction the importance of performing comparative studies in plasmodium-infected mice across distinct time points during mid-pregnancy. The authors should emphasize all biological aspects that endorsed the study. Additionally, it was not clear why days E6.5, E8.5 and E10.5 were chosen to create the models, as well as the choice for B6 mice and Plasmodium chabaudi chabaudi AS plasmodium strain. These points should be taken into consideration to provide a more robust background.

Materials and methods

The statistical analysis applied in the study should be summarized, section is very long.

Results

Comparisons across E6.5, E8.5 and E10.5 groups in relation to mRNA expression of pro- and anti-inflammatory cytokine genes should be carried out to demonstrate which model induce a higher or lower inflammatory profile. This information is important for selecting the most appropriate model to be used in a research. It was not clear why the authors decided to test TNF-/- 313 and TNFRI-/- deficient mice considering that the statistical significance obtained for tnf was less evident in relation to data obtained for infy. Along with, differences were also found for IL-10, Cox-1 and Cox-2 for animals infected on day E8.5, and a higher amount of infy was maintained in animals infected on day E.10.5. Another obscure aspect of this experiment was the fact that only infection occurring at day E8.5 was chosen to be tested. Graphs demonstrating correlation between transcripts and parasitemia are very confusing; it is strongly recommended to remake them, for example, adding the infection day number into the obtained result to facilitate comprehension. Furthermore, some valuable analyses are absent in the work. It would be interesting to insert data related to pups’ viability at weaning (%fetuses), as well as hematocrit change in P. chabaudi chabaudi AS-infected B6 mice. An additional aspect that should be included is a more detailed description of histological analysis to compare if the new models exhibit pathological features associated with human gestational malaria.

Discussion

It would be interesting to discuss more the advantages of this new models compared to the ones already available for testing. Authors should go into greater depth when explaining findings related to Tnf and Il10 transcripts, increased only in the E8.5 infection group. These cytokines are associated with human malaria-induced preterm delivery that was observed in all experimental groups evaluated (E6.5, E8.5 and E10.5). The choice for testing TNF-/-and TNFR 403 -/- mice at E8.5 needs to be better explained. Furthermore, it would be relevant to describe which scientific answers could be answered through the use of each proposed model, highlighting differences between them. Finally, discussion of results that were not presented in the article should be avoided.

Minor comments:

Title:

Line 1: Please correct the word “midgestional” for “midgestational”

Abstract:

Line 30: Please correct the word “midgestional” for “midgestational”

Introduction:

Line 87 to 88: It would be interesting to add information about therapies available to protect Swiss Webster mice against pregnancy loss at midgestation;

Materials and methods:

Line 164: Insert a comma before “as previously described”

Line 198: Remove duplicate parentheses.

Results:

Line 202 to 203: More direct and objective title is needed;

Line 223 to 224: The same title was used;

Line 245 to 248: It would be interesting to add data showing a significant reduction in pup weight (by 0.7546 g, P = 0.0102);

Line 253 to 254: If placenta weight stagnates after E16.5, then we would expect the same value between days E16.5 and E17.5, and a lower value for E15.5 day.

Line 272: Legend from figure 4 better sums up what was written for ‘Placental inflammatory responses do not universally precede preterm delivery’ part. It is strongly recommended to modify this title.

Line 329: Please, remove ‘protective antioxidant defense molecules’ as it seems that they also cause cellular damage.

6. PLOS authors have the option to publish the peer review history of their article (what does this mean?). If published, this will include your full peer review and any attached files.

Reviewer #1: No

Reviewer #2: No

---

## [Author Response · Author response to Decision Letter 0]

22 Feb 2022

Reviewer #1: In this manuscript, the authors describe three murine models of placental malaria, by infecting B6 mice with P. chabaudi chabaudi at embryonic day 6.5, 8.5, or 10.5. The authors observed no intrauterine growth restriction, in contrast to other models of placental malaria, although they do report pre-term delivery. The authors also examined antioxidant and inflammatory gene transcripts, finding variations which are linked to the day of infection. Lastly, the authors have used mice deficient in TNF and its receptor to demonstrate that these do not drive pre-term delivery as a result of infection at embryonic day 8.5.

There are already a number of murine models of placental malaria, and the one described here provides an additional one which can be used to explore responses which result in pre-term delivery.

The statistical analysis methodology is appropriate considering the nature of the study and other recent work in the field.

There are, however, a few areas, which could be improved in the manuscript, listed below.

1. In lines 98 and 99, the authors state that infection during the second trimester is linked to poor outcomes of pregnancy and provide appropriate citations. They also claim that the mouse midgestation period (E6-12) is analogous to the human second trimester, but neither of the citations provided in that sentence support this. The human placenta is fully formed and exposed to maternal blood by the end of the first trimester, whereas the mouse placenta undergoes significant changes between days E6 and E12. These changes include chorioallantoic fusion around E8, and the fetuses are exposed to the full contents of maternal blood by E11.5-12. It is important that the authors clarify if they mean “the middle third of pregnancy” when calling the mouse midgestation and the human second trimester analogous, or if there are specific functional aspects. 

Thank you for pointing this out. Significant changes have been made to the manuscript to be clearer about what other studies have described as the similarities and differences between human and mouse placental development, see lines 112-117. The 2nd half of mouse pregnancy is more closely aligned with the end of the first trimester of human pregnancy, so the manuscript has been modified to reflect that and we have included the appropriate citations.

2. Accounting for this series of changes in the mouse placenta during midgestation, it is pertinent for the authors to explain how the 3 models presented differ with respect to the phase of human disease they aim to recapitulate. 

The reviewer’s comment is appreciated. The manuscript has been significantly modified to more clearly identify how the model described serves to recapitulate human malaria infection during pregnancy. 

3. The authors use precipitous weight loss as an indicator of pre-term delivery (caption for figure 1 and lines 208-210), could this not also be an indicator for resorptions? 

This point is well taken. However, we point out that number of embryo and embryo viability do not differ between infected and uninfected dams when they were euthanized one day prior to anticipated weight loss (S1 Table). This suggests, but doesn’t prove, that the weight loss observed in our model is related to preterm delivery. At these later times in gestation (E16.5-18.5), fetal loss typically presents as, and is used to model, stillbirth, rather than resorptions. Resorptions have been more commonly described as fetal loss during earlier times in gestation (~E7-E10). We’ve updated our manuscript to discuss this and included the appropriate citation. 

4. The histological data used to determine placental parasitemia in Figure 2 would benefit from the inclusion of a representative image showing the difference in maternal and fetal sinusoids in infected mice. 

We appreciate this suggestion and have now included some representative H&E images for each infection group in manuscript, see figure 4.

5. The caption for Figure 3 is difficult to follow e.g “…E6.5 infection group 4IP Dams, 32 placentae…”, Whereas the graph has labels corresponding to sacrifice date, and the number of data points do not appear to correspond (E15.5, 9 points). 

We regret this oversight. The graphs have been updated for clarity.

6. The qPCR method states that pooled samples of minimum 4 placentas were used per dam (lines 167-169), whereas the caption of Figure 4 appears to indicate that each circle represents a placenta (rather than a pool) – this needs to be clarified. 

The figure caption (now Figure 5) has been updated to clarify that each circle represents pooled placentae from individual dams in each group. 

7. For Figure 5, it is not clear why placental parasitemia (measured by histology and averaged, or gene expression in the same samples) was not used instead of, or in addition to maternal parasitemia. 

We have now analyzed the data looking at placental parasitemia as well and updated the figure (now figure 6) and the text to include the significant results. The data reporting placental parasitemia is only for a subset of animals, detected by counting on Giemsa-stained placental sections.

8. In lines 369 to 372, the authors state that these are the first models they know of which do not induce maternal mortality. Models using recrudescence (Sharma et 2016), specifically in multigravid mice (Marinho et al 2009) have also shown this. 

We regret our failure to make our point clearly and appreciate the reviewer pointing this out. I’ve restated this sentence to clarify that our model is one that recapitulates certain aspects of human PM in primigravid hosts that lack immunity or without the need for antimalarial treatment to prevent mortality. This is meant to support the scenario in which malaria causes significant morbidity and impacts pregnancy but is not lethal to the infected mother, if left untreated.

9. The authors do not address the significant physiological differences in the human and mouse placenta within the manuscript, nor do they attempt to explore the potential limitations of their models. Adding this information to the discussion would allow the reader to determine whether the models are appropriate or not. 

This is a very important point, and we appreciate the reviewer making it. We believe the manuscript is much improved with this information included.

Reviewer #2: Major comments:

Introduction:

Overall, it was not clear in the introduction the importance of performing comparative studies in plasmodium-infected mice across distinct time points during mid-pregnancy. The authors should emphasize all biological aspects that endorsed the study. Additionally, it was not clear why days E6.5, E8.5 and E10.5 were chosen to create the models, as well as the choice for B6 mice and Plasmodium chabaudi chabaudi AS plasmodium strain. These points should be taken into consideration to provide a more robust background. 

We thank the reviewer for seeking clarification on our rationale. Because a pregnant woman can be infected during any point during pregnancy, it can be argued that initiating infection at various times points during gestation in a mouse model can provide important information. As we discuss elsewhere in the manuscript (Alicer, in the intro?), use of the B6/PccAS combination offers many benefits: induction of infection earlier in gestation has resulted in confirmation of the importance of several pathologies reported to occur in human placental malaria (cite relevant Poovassery papers, Sarr ox stress and Avery), and this model also allows, as we now show here, induction of infection at various stages of pregnancy that impact embryo and fetal health but do not induce maternal death, as is the case in untreated P. berghei infection. 

Materials and methods

The statistical analysis applied in the study should be summarized, section is very long. 

Done, as requested.

Results

Comparisons across E6.5, E8.5 and E10.5 groups in relation to mRNA expression of pro- and anti-inflammatory cytokine genes should be carried out to demonstrate which model induce a higher or lower inflammatory profile. This information is important for selecting the most appropriate model to be used in a research. 

This is a great suggestion by the reviewer and we have performed univariate and multivariate regression analyses to address this point. See supporting tables 3-10.

It was not clear why the authors decided to test TNF-/- 313 and TNFRI-/- deficient mice considering that the statistical significance obtained for tnf was less evident in relation to data obtained for infy. Along with, differences were also found for IL-10, Cox-1 and Cox-2 for animals infected on day E8.5, and a higher amount of infy was maintained in animals infected on day E.10.5. 

Since TNF has long been associated with adverse pregnancy outcomes in malaria and our lab has previously published work showing that TNF ablation rescues pregnancy compromise, we reasoned that TNF may be important in our new model of malaria-induced preterm delivery. In our previous publication, both TNF and IFNg deficiency rescued pregnancy outcomes in another model of malaria, future studies in our novel model will include infections in IFNg knockout mice. See Poovassery et al, 2009, “Malaria-Induced Murine Pregnancy Failure: Distinct roles of IFNg and TNF” (PubMed 19783682). 

Another obscure aspect of this experiment was the fact that only infection occurring at day E8.5 was chosen to be tested. 

This timepoint was chosen based on the Tnf transcript results shown in Fig. 4.

Graphs demonstrating correlation between transcripts and parasitemia are very confusing; it is strongly recommended to remake them, for example, adding the infection day number into the obtained result to facilitate comprehension. 

This is an excellent point, we have reorganized this figure for clarity. 

Furthermore, some valuable analyses are absent in the work. It would be interesting to insert data related to pups’ viability at weaning (%fetuses), as well as hematocrit change in P. chabaudi chabaudi AS-infected B6 mice. 

Pup viability data is provided in S1 Table and hematocrit change is displayed in S1, S2, and S3 figures for both the observational study and the serial sacrifice study. Live pups were not produced in this study.

An additional aspect that should be included is a more detailed description of histological analysis to compare if the new models exhibit pathological features associated with human gestational malaria. 

As the revised manuscript details, the time points in mouse gestation that were studied here correspond developmentally and functionally to human placental development in the first trimester and beginning of the second trimester. To our knowledge, only a few studies, which we cite, have examined malaria infection this early in gestation and none have reported placental histology. Thus, little is known regarding histological and pathological features of human gestational malaria at this early time in pregnancy. Nonetheless, this time is very important, as it is when maternal blood begins to enter and circulate in the definitive placenta.

We did conduct a histopathological analysis of placental necrosis as an indicator of placental dysfunction, but no differences were found between placentae from infected and uninfected dams. 

Discussion

It would be interesting to discuss more the advantages of this new models compared to the ones already available for testing. 

We appreciate this excellent suggestion to improve our manuscript and have done so. 

Authors should go into greater depth when explaining findings related to Tnf and Il10 transcripts, increased only in the E8.5 infection group. These cytokines are associated with human malaria-induced preterm delivery that was observed in all experimental groups evaluated (E6.5, E8.5 and E10.5). The choice for testing TNF-/-and TNFR 403 -/- mice at E8.5 needs to be better explained. Furthermore, it would be relevant to describe which scientific answers could be answered through the use of each proposed model, highlighting differences between them. Finally, discussion of results that were not presented in the article should be avoided.

All excellent points made by the reviewer. We have clarified our use of TNF mice in lines 353-355. We have also speculated on some scientific answers that this proposed model may help to address in the discussion, lines 510-522.

Minor comments:

Title:

Line 1: Please correct the word “midgestional” for “midgestational” 

Done as requested.

Abstract:

Line 30: Please correct the word “midgestional” for “midgestational” 

Done as requested.

Introduction:

Line 87 to 88: It would be interesting to add information about therapies available to protect Swiss Webster mice against pregnancy loss at midgestation; 

We are currently conducting studies to characterize this and our findings will be reported in a separate paper.

Materials and methods:

Line 164: Insert a comma before “as previously described” 

Done as requested. 

Line 198: Remove duplicate parentheses. 

Done as requested.

Results:

Line 202 to 203: More direct and objective title is needed; 

Done as requested.

Line 223 to 224: The same title was used;

Clarified, as requested. 

Line 245 to 248: It would be interesting to add data showing a significant reduction in pup weight (by 0.7546 g, P = 0.0102); 

This reduction was only observed in the context of litter size in the E10.5 group, so as number of viable pups increased, pup weights decreased. This phenomenon has been described before and the reference is provided.

Line 253 to 254: If placenta weight stagnates after E16.5, then we would expect the same value between days E16.5 and E17.5, and a lower value for E15.5 day.

We understand the confusion here and opted to remove this interpretation from the manuscript as it is not part of the core paper. 

Line 272: Legend from figure 4 better sums up what was written for ‘Placental inflammatory responses do not universally precede preterm delivery’ part. It is strongly recommended to modify this title. 

Title has been removed to adhere with PLOS One formatting standards. 

Line 329: Please, remove ‘protective antioxidant defense molecules’ as it seems that they also cause cellular damage. 

Done, as requested.

---

## [Decision Letter · Decision Letter 1]

8 Mar 2022

A novel murine model of post-implantation malaria-induced preterm birth

PONE-D-21-23277R1

Dear Dr. Andrew,

We’re pleased to inform you that your manuscript has been judged scientifically suitable for publication and will be formally accepted for publication once it meets all outstanding technical requirements.

Kind regards,

Claudio Romero Farias Marinho, Ph.D.

Academic Editor

PLOS ONE

Additional Editor Comments (optional):

Reviewers' comments:

Reviewer's Responses to Questions

**Comments to the Author**

1. If the authors have adequately addressed your comments raised in a previous round of review and you feel that this manuscript is now acceptable for publication, you may indicate that here to bypass the “Comments to the Author” section, enter your conflict of interest statement in the “Confidential to Editor” section, and submit your "Accept" recommendation.

Reviewer #2: All comments have been addressed

2. Is the manuscript technically sound, and do the data support the conclusions?

Reviewer #2: Yes

3. Has the statistical analysis been performed appropriately and rigorously? 

Reviewer #2: Yes

4. Have the authors made all data underlying the findings in their manuscript fully available?

Reviewer #2: Yes

5. Is the manuscript presented in an intelligible fashion and written in standard English?

Reviewer #2: Yes

6. Review Comments to the Author

Reviewer #2: (No Response)

7. PLOS authors have the option to publish the peer review history of their article (what does this mean?). If published, this will include your full peer review and any attached files.

Reviewer #2: No

---

## [Editor Report · Acceptance letter]

11 Mar 2022

PONE-D-21-23277R1 

A novel murine model of post-implantation malaria-induced preterm birth 

Dear Dr. Andrew:

I'm pleased to inform you that your manuscript has been deemed suitable for publication in PLOS ONE. Congratulations! Your manuscript is now with our production department. 

Kind regards, 

on behalf of

Dr. Claudio Romero Farias Marinho 

Academic Editor

PLOS ONE